# A Rare Case of Peripheral Osteoma of the Alveolar Bone of the Maxilla in a 13-Year-Old Boy

**DOI:** 10.3390/jcm13237187

**Published:** 2024-11-27

**Authors:** Ivana Gušić, Marija Stojilković, Jelena Mirnić, Tanja Veljović, Milanko Đurić

**Affiliations:** 1Department of Dental Medicine, Faculty of Medicine, University of Novi Sad, Hajduk Veljkova 3, 21000 Novi Sad, Serbia; marija.stojilkovic@mf.uns.ac.rs (M.S.); jelena.mirnic@mf.uns.ac.rs (J.M.); tanja.veljovic@mf.uns.ac.rs (T.V.); milanko.djuric@mf.uns.ac.rs (M.Đ.); 2Department for Periodontology and Oral Medicine, Dentistry Clinic of Vojvodina, Hajduk Veljkova 12, 21000 Novi Sad, Serbia

**Keywords:** osteoma, alveolar processus, maxilla, bone tissue, pediatric, follow-up

## Abstract

**Background:** This report aims to augment the presently limited knowledge on the characteristics of jawbone osteomas in children by presenting an exceptionally rare case of this tumor located on the buccal aspect of the alveolar process of the maxilla in a 13-year-old boy. **Methods:** A well-defined, painless, bony, hard, spherical enlargement on the maxillary alveolar ridge was identified and thoroughly evaluated through clinical examination, panoramic radiographs, CBCT (Cone Beam Computed Tomography) scans, and histopathological analysis. The tumor was surgically removed, and the patient participated in postoperative clinical follow-ups for eight years. **Results:** Based on the clinical characteristics and CBCT scan findings, a jawbone tumor was suspected. After histopathological analysis, the definitive diagnosis was a peripheral trabecular osteoma. There were no signs of tumor recurrence during the postoperative follow-up period. **Conclusions:** This report presents the youngest documented case of peripheral osteoma in the maxillary alveolar ridge, and highlights the need to consider this rare lesion in the differential diagnosis of similar pathological changes in this region, even in pediatric patients. The absence of clinical signs of recurrence over eight years of follow-up underscores the long-term stability and favorable prognosis of peripheral jawbone osteoma in children.

## 1. Introduction

Osteomas are rare benign bone neoplasms that almost exclusively affect the craniofacial skeleton [1]. They most commonly arise from periosteum activation, forming peripheral osteomas that grow on the bone surface, while, less frequently, they originate from endosteum activation, forming central osteomas enclosed within the bone tissue. Peripheral osteomas are the most prevalent type on the craniofacial skeleton [2]. They are typically detected in individuals over 30 years of age [1,2,3], with no significant gender predilection [1,2,4,5]. These tumors are most commonly found in the paranasal sinuses and areas of the mandible exposed to trauma [2,5,6,7]. However, peripheral osteomas in the upper jaw, excluding the maxillary sinus, are extremely rare, with only 15 cases reported to date [2,4,5,8,9,10,11,12,13,14,15,16,17,18].

Peripheral osteomas generally grow slowly and are often asymptomatic, though they can interfere with normal function or exert pressure on surrounding structures, leading to symptoms depending on their location. Most asymptomatic osteomas are discovered accidentally during clinical examination or analysis of radiographs acquired for other reasons, which is often sufficient for a definitive diagnosis. If conventional radiographs do not provide sufficient information for a definitive diagnosis or treatment planning, computed tomography (CT) is recommended. Cone Beam CT (CBCT), which offers a lower radiation dose, can serve as an alternative to traditional CT [1,19,20]. While histopathological analysis is typically not required, it may be necessary in uncertain cases. Pathohistologically, osteomas appear as either lamellar compact or cancellous bone [21]. Surgical resection using a minimally invasive approach is indicated if the tumor causes functional impairment or aesthetic concerns [1].

In pediatric patients, multiple jaw osteomas are commonly associated with Gardner syndrome [22,23]. However, solitary osteomas in children are exceedingly rare, with only a few cases reported in the literature [16,17]. This report presents a rare case of a peripheral cancellous osteoma in a 13-year-old boy located on the alveolar process of the maxilla. This case represents the youngest patient reported and is the 16th documented case of a peripheral osteoma localized on the maxillary alveolar ridge. The article commences with the clinical evaluation, followed by the orthopantomogram and CBCT radiogram analyses, after which treatment by surgical removal, histopathological findings, and 8-year postoperative follow-ups are described.

The aim is to present an unusual case from clinical practice in order to increase the awareness of the possibility of osteoma on the alveolar part of the maxilla in pediatric age, and to contribute to the knowledge about the characteristics of these bone tumors in children.

## 2. Case Presentation

A 13-year-old boy, accompanied by his parents, presented to the Department of Periodontology and Oral Medicine, Dentistry Clinic of Vojvodina, Novi Sad, Serbia, owing to a referral due to a painless enlargement in the upper maxillary region. The growth was noticed by his pediatric dentist during an appointment for the restoration of carious teeth. Neither the patient nor his parents could recall when the enlargement first appeared or how quickly it expanded in size. However, according to the mother, the dentist did not note any changes during the last visit two years prior. Apart from difficulty brushing his teeth, the boy had no subjective complaints. They denied the habits of teeth grinding or jaw clenching, and denied any physical trauma in the affected region. The boy was in good general health and had never undergone any surgical procedures. Neither the patient nor his immediate family members have had any medical conditions relevant to the described case.

Clinical examination revealed a clearly circumscribed spheric enlargement of hard bone-like consistency that was insensitive to palpation and extended from the mesial surface of the first premolar to the mesial surface of the first molar. In the apico-coronal aspect, the growth expanded from the upper buccal fornix to the occlusal surface of both upper right premolars, covering their buccal surfaces to the cusp tips (Figure 1).

The involved teeth were healthy, properly aligned in the dental arch, and reacted normally to the electro-test for pulp vitality. The mucous membrane above the lesion seemed tense and somewhat paler compared to the surrounding gingiva. The lesion was not visible from the palatal aspect. Dental biofilm accumulations were present on adjacent teeth, which was consistent with the otherwise low level of oral hygiene in this patient. There were no clinical signs of occlusal trauma. The described lesion was not visible on the classical orthopantomographic radiograph. A normal bone pattern was evident, without evidence of pathological changes associated with the teeth or maxillary sinuses (Figure 2). The differential diagnosis considered included peripherial osteoma, ossifying fibroma, osteoid osteoma, osteoblastoma, and osteosarcoma.

Due to the bone-hard consistency of the protuberance, a CBCT of the affected region was performed to further aid in diagnosis. The radiograph revealed an exophytic mushroom-like shadow of varying radiological density, which was in places similar to normal bone. It extended in the coronal and buccal directions, manifesting as a continuity of the cortical and spongy structure of the thickened marginal part of the alveolar bone (Figure 3). There were no similar enlargements elsewhere in the jawbones.

The growth was confirmed as benign, and the patient was scheduled for surgical removal under local anesthesia.

After anesthetizing, the bony protuberance was fully exposed by elevating the mucoperiosteal flap with a retractor. During blunt dissection of the flap, a portion of the submucosal bony structure was crushed into smaller fragments. The remaining bony surface was uneven and unchanged in color compared to the surrounding bone. The remnants of the bony enlargement were removed using a bur mounted on a handpiece, with constant cooling, by cutting through the area of the wide base. After rounding the sharp contours, the affected region was covered with coronary mobilized alveolar mucosa and closed with single sutures (Figure 4).

Aside from an unusually pronounced gag reflex for this region (which complicated the procedure and photography during the operation), the surgical course was uneventful, with no complications in the postoperative period. The sutures were removed seven days after the procedure (Figure 5).

A portion of the bony prominence was sent for pathological analysis. Histopathologically, the lesion was characterized as a bony exostosis/hyperostosis. The specimen’s structure consisted of uneven bony trabeculae separated by connective tissue with dilated blood vessels (Figure 6a,b). Based on the clinical features and these observations, the final diagnosis was peripheral trabecular osteoma.

Considering the unclear lesion etiology, the patient was scheduled for semi-annual check-ups. During the 8-year follow-up period, there were no signs of recurrence or emergence of new jawbone osteomas (Figure 7). Since the patient’s parents did not consent to a colonoscopy, the boy has been clinically monitored for Gardner syndrome. During follow-up visits, a detailed anamnesis was carefully conducted to check for the development of symptoms, such as the appearance of new bone tumors or gastrointestinal tract symptoms.

## 3. Discussion

This report describes an unusual case involving a prominent bony mass on the buccal aspect of the alveolar ridge of the maxilla in a 13-year-old child characterized as peripheral trabecular osteoma. According to the World Health Organization’s *International Classification of Diseases* (WHO *ICD*) in Dentistry, osteomas are benign neoplasms of bone tissue [24] usually diagnosed in adults. The average age at the time of their emergence varies significantly across populations and geographic areas [2,5,11]. For example, in the largest retrospective study conducted in Spain, which included 106 patients with osteomas of the craniofacial region, subjects were 50 years old on average [2], whereas in a study involving 35 Turkish patients with craniofacial osteoma localization, the average age was 29.4 years [5], and a Korean sample comprising 18 patients had a mean age of 42.2 years [21]. These findings highlight substantial geographic and demographic variability, suggesting that environmental or genetic factors might influence the timing of osteoma development. However, the underlying reasons for these differences remain unclear. This variability raises questions about whether certain populations are predisposed to earlier or later onset due to lifestyle or genetic predispositions, or if health care access affects the timing of when a diagnosis is made. In the context of this case, the early onset in this 13-year-old patient might point to unique etiological factors that distinguish solitary pediatric osteomas from those typically observed in adults. This case underscores the importance of expanding research into osteoma development patterns to better understand age-related trends and their clinical implications.

In general, primary lesions in the jawbones are not common in children and osteomas have been reported in only seven pediatric patients [5,16,25,26,27]. In Sayan et al.‘s study, among 35 osteomas, only one case was recorded in a pediatric age group, which was in a 14-year-old child [5]. Available research on the characteristics of jawbone tumors in children indicate that solitary osteomas are rarely part of this pathology. Indeed, Krasnean et al. identified only one osteoma in a sample of 82 jawbone tumors in children in Romania [25], whereas only 3 of the 104 pediatric patients with jawbone lesions in Massachusetts had osteomas [27]. On the other hand, a retrospective analysis of 100 benign pediatric jawbone lesions according to the new WHO *ICD* guidelines did not reveal a single osteoma [28].

While jaw osteomas in pediatric patients are rare, the majority of reported cases pertain to multiple bone lesions related to Gardner syndrome [22,23], which is typically diagnosed between ages 13 and 31 [22]. Gardner syndrome is an inherited condition characterized by intestinal polyposis, multiple osteomas, and soft tissue tumors. In individuals affected by this condition, intestinal polyps represent premalignant lesions which, according to some authors, always become malignant [22]. Sometimes, the incidental finding of an osteoma on the jaw in children is the first clinical sign of a serious condition, so dentists can play a crucial role in the early recognition of this otherwise rare syndrome.

The exact cause of osteomas remains unclear, and various theories have been proposed regarding their development. Their slow and asymptomatic growth is not characteristic of neoplastic processes. Furthermore, they are most commonly located in areas exposed to chronic inflammation or mechanical trauma, leading many to consider them to be reactive lesions [29,30,31,32,33,34]. According to the inflammatory theory, the formation of peripheral osteomas is induced by chronic infection and inflammation in close proximity to the bone surface, which initiates the proliferation of osteogenic cells associated with the periosteum and eventually leads to tumor development. These postulates align with the frequent appearance of osteomas in the paranasal sinuses, where they are usually accompanied by chronic sinusitis [35].

In our case, the patient had generalized marginal gingivitis due to poor oral hygiene, so it is possible that there was initially gingival inflammation at the site of the affected premolars. However, linking this type of infection to periosteal irritation and subsequent bone proliferation is challenging. Given the frequency of gingivitis, particularly during puberty, one would expect the development of osteomas in this region if there were a connection to this type of infection. Injury sites such as the angulus and lower edge of the mandibular body are the second-most common osteoma localizations [30,31,32]. This has led some authors to consider mechanical trauma as a potential cause of reactive bone enlargement and osteoma development [32].

The localization of the protuberance in our patient is not considered a typical site of mechanical injury. However, in several published cases of maxillary alveolar osteomas, injury was hypothesized to be the initiating cause of the lesion [32]. Although the patient and his mother categorically denied any trauma, the osteoma may have developed as a result of a minor injury that went unnoticed.

According to developmental or embryological theory, osteomas are developmental anomalies that arise from embryological cartilaginous rests or from persistent embryological periosteum. Although this hypothesis cannot explain the occurrence of most osteomas in adults, it is applicable in children and adolescents, as shown by Dalambiras et al., who reported on a pediatric case involving osteoma localized near the premaxilla–maxilla junction [26]. In our patient, the tumor grew in the region between the upper premolars, making its origin incompatible with this theory.

The clinical implications of jawbone osteomas can vary depending on their location and size. In particular, the neoplastic nature of jawbone osteomas is reflected in their potential to compromise tooth eruption [26], cause their displacement [36], or lead to tooth root resorption in the affected region [37]. However, in our case, the bony mass did not exhibit any of the aforementioned characteristics of neoplastic growth. The tumor arose from the marginal part of the alveolar bone outward and did not compress the teeth it obviously passively rested upon. The teeth remained in their preserved position and vitality. Similar non-invasive characteristics were exhibited by the trabecular osteoma of the palatal aspect of the alveolar ridge described in a 17-year-old girl. [16]. In this case, although the tumor covered most of the palate, it did not affect the position and vitality of the teeth at the site of the bony protrusion’s base. On the other hand, in the case of a 16-year-old female patient presented by Dalambiras et al., a peripheral cortico-type osteoma clearly compromised tooth eruption [26]. Even more aggressive behavior was observed in multiple osteomas of the jaws due to Gardner syndrome in a 15-year-old girl [23]. It can be assumed that the diagnosis of jaw osteoma encompasses tumors of varying invasive potential, or it may involve bone proliferations of different natures. It is possible that the spongy type of jaw osteoma primarily grows into free space and does not compress adjacent structures. According to Kaplan et al., some lesions classified as osteomas should rather be classified as exostoses, whereby those caused by trauma should be described as “parosteal osseous hyperplasia” and the term “osteoma” should be reserved for true benign neoplastic lesions [6].

Osteomas are generally not difficult to diagnose, and in most cases a clinical examination and analysis of a conventional radiograph are sufficient. Histopathological analysis is only necessary in cases where a definitive diagnosis cannot be reached. However, all cases of peripheral osteomas of the jaws published in extant literature were clearly visible on a classic two-dimensional radiograph. This can be explained by the fact that most of these tumors were of the cortical type and consisted of abnormally dense bone derived from the periosteum or bone marrow, thereby producing a convincing opacity on an X-ray image. The trabecular structure noted in our young patient is a less common feature of such osteomas [20]. In a recent report pertaining to a 70-year-old patient, multiple osteomas of the alveolar bone of both jaws were described, and were linked to mechanical trauma caused by dental implant placement. In this case, bone tumors in the maxilla had a trabecular structure, while all osteomas of the alveolar part of the mandible were of the cortical type [32]. It is possible that the greater proportion of cancellous bone in the maxilla compared to the mandible influences the differences in the structures of osteomas in the upper and lower jaws. Similar radiological characteristics, namely, CBCT findings, as in our case were exhibited by a peripheral, trabecular osteoma of the palatal aspect of the alveolar bone reported by Omezzine et al. [16].

In our case, the peripheral trabecular osteoma diagnosis was based on its clinical characteristics (a painless, mushroom-like, bony prominence) as well as its microscopic features, which were consistent with exostosis. The terms “exostoses”, “tori”, and “osteomas” are often used interchangeably in the scientific literature [21,33]. All of these structures are benign, localized, peripheral bone enlargements that do not differ histologically [38]. The designation of prominence is usually determined by localization and clinical characteristics. Tori are typically located at the midline of the hard palate and the lingual surfaces of the body of the mandible. On the other hand, exostoses are considered non-pathological bone enlargements. Buccal exostoses typically affect the jaws, especially the alveolar ridge of the maxilla, almost always bilaterally in the projection of the roots of the back teeth [39,40]. While toruses and exostoses are usually diagnosed in young adults, in our patient, the age of onset, growth localization, and the clinical manifestation did not exhibit jaw torus or buccal alveolar exostosis characteristics. In clinical practice, non-pathological and neoplastic bone enlargements are distinguished by their size and growth characteristics [1,41]. Unlike exostoses and jaw toruses, osteomas tend to increase in size continually, albeit slowly, with notable differences in their growth pattern [1]. On average, these tumors increase by 0.43–1.73 mm in diameter per year [33,42], but much more rapid enlargement has been noted, coinciding with the period of intense osteogenesis. In boys, bones of the orofacial region grow most rapidly during the ages of 12–14 [43], which may explain the size of the bony protuberance in our 13-year-old patient that, by the time he was seen at our clinic, had already extended from the fornix to the tips of the buccal cusps of both upper premolars. The lesion’s shape and dimensions, as well as the boy’s age, were indicative of its neoplastic origin. However, the tumor’s unusual location, and the fact that it was not visible on the initial X-ray, raised additional questions about its nature.

Peripheral ossifying fibroma was initially considered as part of the differential diagnosis. Similar to the presented case, this type of fibroma commonly arises during the second decade of life, typically originates in the interdental region, and is usually not visible on conventional radiographs. However, peripheral ossifying fibroma is a reactive tumor of the gingiva that most commonly occurs in females, typically in the incisor and canine region [44,45]. Depending on the degree of ossification, it may have a soft or firm elastic consistency. The affected gingiva is usually pink or red and is sometimes ulcerated [45]. In our patient, the tumor had a bone-hard consistency, and the overlying mucosa appeared pale and tense. On CT images, peripheral ossifying fibroma manifests as radiopaque diffuse calcifications in a shadow of soft tissues. While the underlying bone surface typically remains unchanged, erosion and even destruction may occur due to tumor compression [46]. In our patient, the CBCT scan revealed buccal bone proliferation, which was confirmed intraoperatively, requiring a burr for lesion removal, thus ruling out a tumor of soft-tissue origin. Accordingly, peripheral ossifying fibroma was excluded.

During the differential diagnosis, benign bone tumors such as osteoid osteoma and osteoblastoma were also considered. These tumors are exceedingly rare in the maxillofacial region and are more commonly observed in the mandible [47]. Moreover, both tumors are typically painful, whereas our patient was symptomless. In osteoblastoma, pain increases over time and does not respond to NSAIDs, while in osteoid osteoma it worsens at night but is relieved by NSAIDs. Radiographically, both tumors are characterized as mixed-density lesions, whereby osteoid osteoma presents as a small radiolucent nidus surrounded by a dense sclerotic zone, while osteoblastoma often expands and erodes surrounding bone. Histopathologically, both tumors show abnormal osteoblast proliferation forming osteoid and woven bone on a fibrous connective tissue stroma, whereas our patient had a normal bone structure. Thus, as none of the criteria for osteoblastoma and osteoid osteoma were met, these diagnoses were ruled out as well [47].

There was significant concern that the observed pathological lesion might represent a malignant condition, such as a jawbone sarcoma. Jaw osteosarcoma can occur at any age, with one peak incidence between ages 10 and 14, corresponding to our patient’s age. CBCT and histopathological findings in our case strongly suggested a benign etiology of the tumor. However, osteosarcoma is known for presenting with nonspecific or even misleading clinical radiographic features, which frequently contribute to diagnostic delays. Moreover, when small or fragmented biopsy samples are obtained, distinguishing between low-grade osteosarcoma and osteoma becomes notably challenging, if not impossible. To date, there have been no reported cases of malignant transformation associated with peripheral osteomas. However, existing literature does document instances in which pediatric osteosarcomas were initially misidentified as osteomas [48].

Although osteomas are predominantly benign and exhibit a low recurrence rate, there are notable exceptions where recurrence may occur, particularly when complete surgical excision has not been successfully achieved [49]. Recognizing these diagnostic complexities, we opted for prolonged clinical and radiographic follow-up in this patient, continuing for several years post surgical excision of the osteoma to ensure timely identification of any potential recurrence or alternative pathology. During the eight-year postoperative period, the patient underwent detailed clinical examinations at six-month intervals, which included visual inspection and palpation of the alveolar ridge at the site of the previous tumor, as well as assessment of the vitality of the affected teeth. Three years after surgery, a panoramic radiograph was performed, showing no pathological changes, after which clinical follow-up was continued exclusively. In the postoperative period of eight years, no signs of recurrence or new bone tumors in the jaw area were observed. Additionally, the patient did not report any gastrointestinal symptoms suggestive of Gardner syndrome.

## 4. Conclusions

The etiopathogenesis and clinical characteristics of jaw osteomas involve considerable uncertainty and variability. In extant literature, there are very few cases of this type of jawbone tumor in children. Thus, there is an evident need for new clinical case reports and retrospective studies focusing on the pediatric population, as a detailed description of the pathogenesis, as well as clinical, radiological, and histopathological characteristics of jaw osteomas would facilitate further comparative analyses, helping increase our knowledge of benign alveolar bone enlargements in pediatric patients.

## Figures and Tables

**Figure 1 jcm-13-07187-f001:**
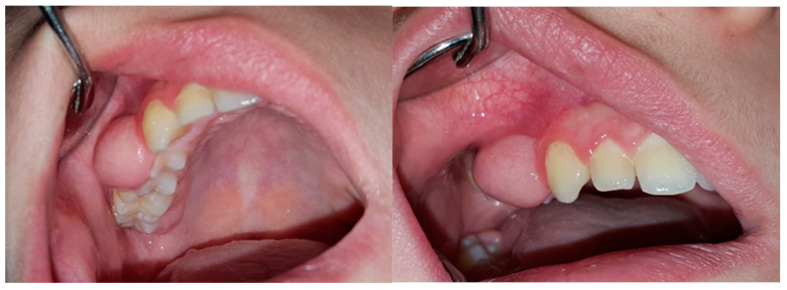
Intraoral views of the lesion.

**Figure 2 jcm-13-07187-f002:**
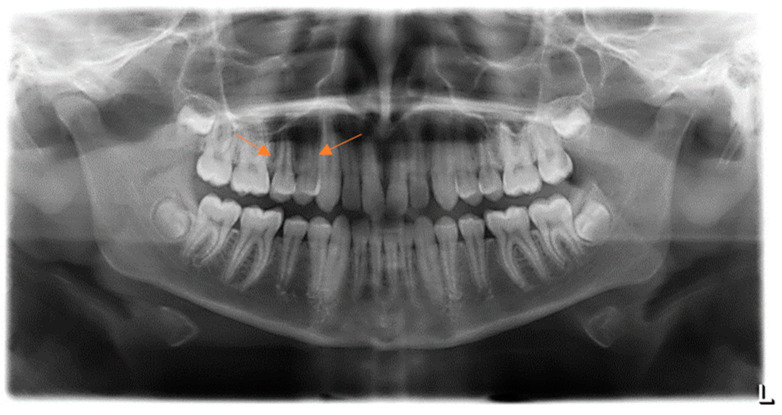
Panoramic radiography at the first visit. The arrows indicate the location of the osteoma.

**Figure 3 jcm-13-07187-f003:**
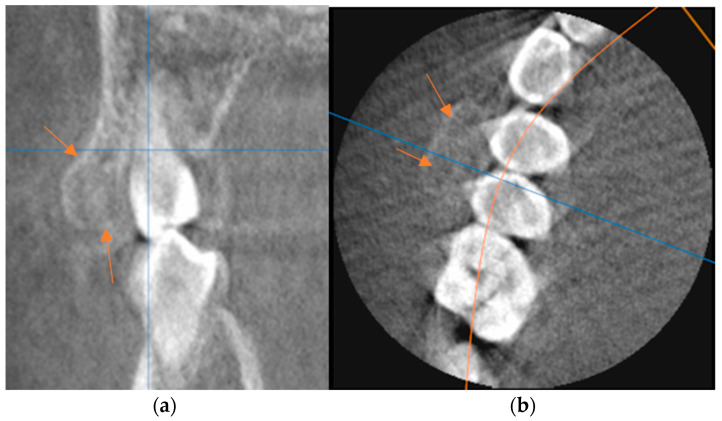
Preoperative (**a**) sagittal and (**b**) axial images from CBCT showing the proliferated bone mass on the buccal bone of the maxillary premolars, with an external cortical layer and an internal radiolucent marrow space. The arrows indicate the location of the osteoma.

**Figure 4 jcm-13-07187-f004:**
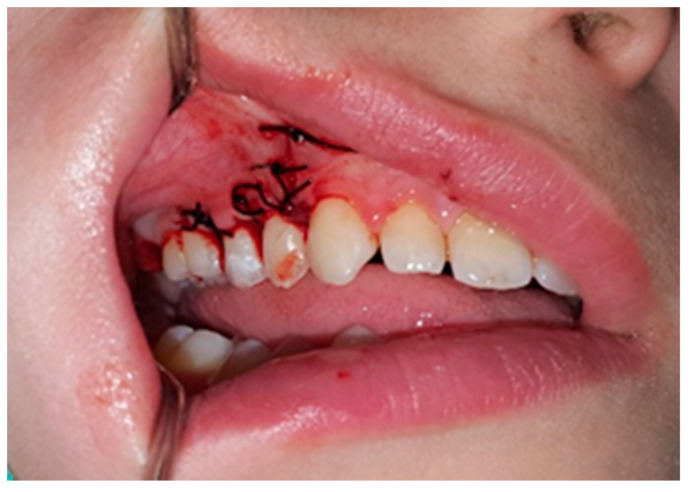
Postoperative image of the patient showing sutures in the surgical site.

**Figure 5 jcm-13-07187-f005:**
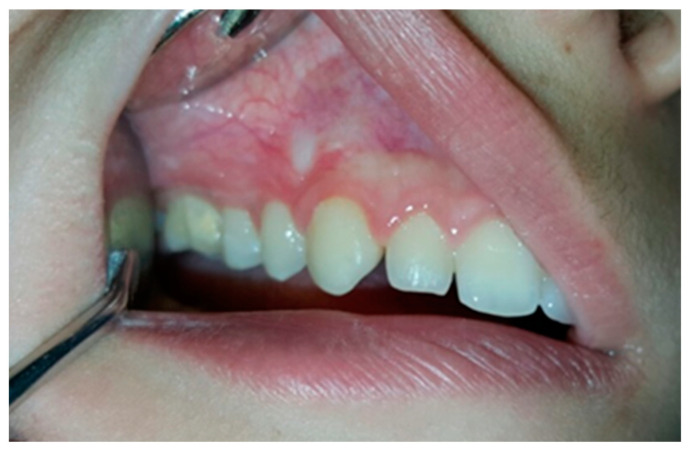
Post-treatment healing after 10 days.

**Figure 6 jcm-13-07187-f006:**
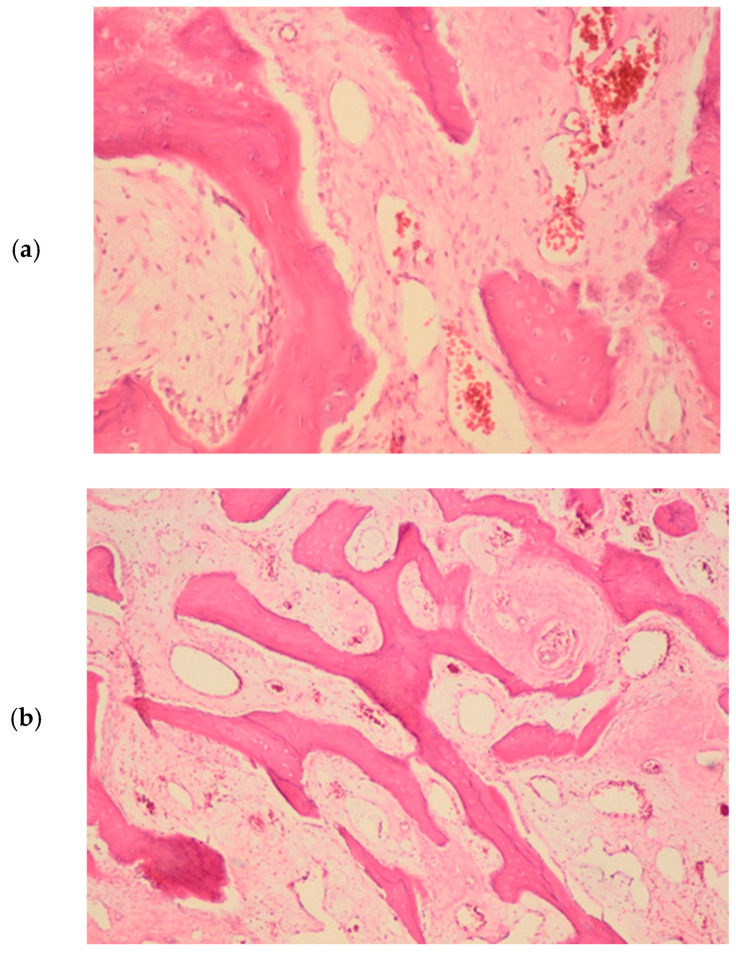
Histologic features (H–E stain) (**a**) Bone with uneven bone beds interspersed with connective tissue with dilated blood vessels (HE × 40) (**b**) Bone fragment with preserved osteoclasts, osteoblasts, and osteocytes of appropriate number and morphology. The bone beds are separated by connective tissue with dilated blood vessels (HE × 100).

**Figure 7 jcm-13-07187-f007:**
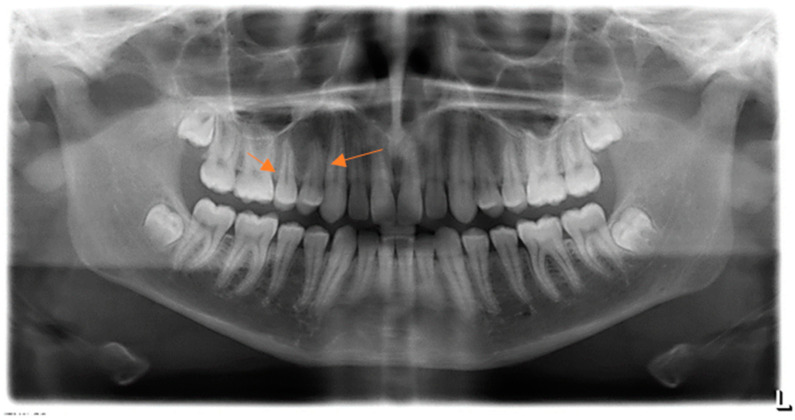
Panoramic radiograph of the patient taken 3 years postoperatively. The arrows indicate the location of the osteoma.

## Data Availability

No new data were created or analyzed in this study. Data sharing is not applicable to this article.

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
