# Peer review of "A Rare Case of Peripheral Osteoma of the Alveolar Bone of the Maxilla in a 13-Year-Old Boy"

_jcm, 2024, doi:10.3390/jcm13237187_

Round 1
Reviewer 1 Report
Comments and Suggestions for Authors
Changes suggestion for abstract.
- Conciseness: Removed some redundancy and streamlined the language (e.g., changing "peripheral trabecular osteoma diagnosis was made" to "diagnosis was peripheral trabecular osteoma").
- Clarity: Combined some sentences for better flow and clarity, such as describing the CBCT scan findings and postoperative follow-up together in a single sentence.
- Grammar: Corrected the repetition of "on the" in the phrase " on the alveolar ridge."
changes suggestion for Discussion section
- Age-related Comparisons: While the authors reference studies from different geographic regions and populations that note varying average ages of osteoma presentation, the significance of these differences remains underexplored. How does this variability impact the general understanding of osteoma development? Is there any pattern or trend that could link this particular case to these broader findings?
- Differential Diagnosis Discussion: The differential diagnosis section is critical, as it rules out conditions like peripheral ossifying fibroma and jaw osteosarcoma. However, the article could strengthen this part by providing a more detailed explanation of why certain conditions were excluded based on radiographic findings, clinical presentation, and histopathological results. For instance, why were peripheral ossifying fibroma and osteoid osteoma ruled out with such confidence despite some clinical overlap?
- Follow-up and Monitoring: The article concludes by emphasizing long-term follow-up post-surgery to monitor for recurrence or malignancy, but it would benefit from more details about the specific protocols for this follow-up. A brief explanation of what imaging techniques or clinical assessments were used over the period would offer practical insight into the management of such cases and inform readers on best practices.
- General Structure and Flow: The article could benefit from improved structure, with a more logical flow between sections. For example, the initial discussion of terminologies might be better placed after the case description, leading into the differentiation of the osteoma from other bone enlargements. Additionally, more transitional sentences between paragraphs could help the reader follow the complex discussion.
In conclusion, this report provides valuable insights into a rare pediatric case of peripheral trabecular osteoma. However, addressing the aforementioned critiques—by improving the structure, providing clearer etiological hypotheses, enhancing the differential diagnosis discussion, and including visual documentation—would significantly strengthen the clinical relevance and clarity of the paper.

Author Response
Comment 1: Abstract Conciseness: Removed some redundancy and streamlined the language (e.g., changing "peripheral trabecular osteoma diagnosis was made" to "diagnosis was peripheral trabecular osteoma"). Clarity: Combined some sentences for better flow and clarity, such as describing the CBCT scan findings and postoperative follow-up together in a single sentence. Grammar: Corrected the repetition of "on the" in the phrase " on the alveolar ridge."
Response 1: Based on editor suggestions, we have revised the abstract to follow the structured format with distinct sections for Purpose, Methods, Results, and Conclusions. We have also streamlined the language to avoid redundancy and improved clarity, while ensuring that all key details and findings are retained. We hope the revised version aligns better with the journal's expectations.
Comment 2: Changes suggestion for Discussion section , Age-related Comparisons: While the authors reference studies from different geographic regions and populations that note varying average ages of osteoma presentation, the significance of these differences remains underexplored. How does this variability impact the general understanding of osteoma development? Is there any pattern or trend that could link this particular case to these broader findings?
Response 2: In response to your comment regarding the geographic and demographic variability of osteomas, we have expanded the discussion section to incorporate a more thorough analysis of these aspects. We highlighted the possibility that genetic predispositions, or access to healthcare could contribute to the earlier or later onset of symptoms or the timing of diagnosis in certain populations. We believe these additions provide a more comprehensive view on the topic, underscoring the need for further research to understand the age-related trends and their clinical implications.
Comment 3: Differential Diagnosis Discussion: The differential diagnosis section is critical, as it rules out conditions like peripheral ossifying fibroma and jaw osteosarcoma. However, the article could strengthen this part by providing a more detailed explanation of why certain conditions were excluded based on radiographic findings, clinical presentation, and histopathological results. For instance, why were peripheral ossifying fibroma and osteoid osteoma ruled out with such confidence despite some clinical overlap?
Response 3: In response to your suggestion, we have expanded the discussion in the differential diagnosis section to clarify why certain conditions were excluded. We have provided detailed reasoning based on the clinical presentation, radiographic findings, and histopathological results.
Comment 4: Follow-up and Monitoring: The article concludes by emphasizing long-term follow-up post-surgery to monitor for recurrence or malignancy, but it would benefit from more details about the specific protocols for this follow-up. A brief explanation of what imaging techniques or clinical assessments were used over the period would offer practical insight into the management of such cases and inform readers on best practices.
Response 4: We have added more detailed information on the protocol used during the postoperative follow-up. We hope this added detail addresses your comment.
Comment 5: General Structure and Flow: The article could benefit from improved structure, with a more logical flow between sections. For example, the initial discussion of terminologies might be better placed after the case description, leading into the differentiation of the osteoma from other bone enlargements. Additionally, more transitional sentences between paragraphs could help the reader follow the complex discussion.
Response 5: We appreciate your suggestion to improve the logical progression of the sections. In response, we have revised the manuscript to move the discussion of terminologies to a later section. Additionally, we have added few transitional sentences between paragraphs to enhance the readability and ensure a more coherent flow throughout the discussion. We hope that these adjustments address your concerns, and we are grateful for your constructive feedback.

Reviewer 2 Report
Comments and Suggestions for Authors
this report present a rare case of osteoma in maxillar area in 13 years old boy
some point to be modified
line 14: change pathohistological >>> histopathological
line 15 : follow-up for 8 years (add if no recurrence or recurrence)
line 19 : CBCT scan and pathohistological analysis (define CBCT) and change to histopathological analysis
introduction : very long for a case report, please shorten it to 3-4 paragraph maximally
case presentation : add if previous surgical/medical /family history present
line 77-79 : The article commences with the clinical evaluation, followed by the orthopantomogram, CBCT radiogram analyses, and pathohistological findings, after which treatment by surgical removal and 6-year postoperative follow-up are described.
>> it was 6 year or 8 years ?
figures : add arrows describing area of osteoma on CBCT and x-rays
was there any further investigation for gardner syndrome ?
Add potential differential diagnosis in case presentation and discussion
Author Response
Comment 1: >line 14: change pathohistological >>> histopathological
Response 1: Thank you for pointing out this error. Throughout the text, the term "histopathological" has been used instead of "pathohistological." The changes in the text are highlighted in yellow.
Comment 2 : line 15: follow-up for 8 years (add if no recurrence or recurrence)
Response 2: Following the editor's suggestion, the abstract has been revised to be structured into sections. The part addressing the absence of signs of recurrence is now included in the Results section of the abstract. We hope this meets your expectations.
Comment 3:>line 19 : CBCT scan and pathohistological analysis (define CBCT) and change to histopathological analysis
Response 3: The abbreviation CBCT is now explained in parentheses at its first occurrence in the abstract.
Comment 4: introduction : very long for a case report, please shorten it to 3-4 paragraph maximally
Response 4: Thank you for pointing out that the introduction was too lengthy. We have made an effort to significantly shorten this section, condensing it into a few key paragraphs. The first paragraph, which generally discussed jaw tumors in children, as well as information on prevalence, pathogenesis, and detailed descriptions of the clinical presentation of peripheral osteomas, has been removed. This content was simply deleted, but we are unsure if it should have been handled differently to indicate the changes made. If this is the case, we will submit a revised version where the removed sections are clearly marked.
Comment 5: case presentation : add if previous surgical/medical /family history present
Response 5: Information about the patient’s previous surgeries and medical history, as well as that of his family, has been added to the case presentation section and is highlighted in yellow in the text.
Comment 6: line 77-79 : The article commences with the clinical evaluation, followed by the orthopantomogram, CBCT radiogram analyses, and pathohistological findings, after which treatment by surgical removal and 6-year postoperative follow-up are described. >> it was 6 year or 8 years ?
Response 6: Thank you for pointing out this error. Initially, we intended to publish this paper two years ago when we began drafting the manuscript. Later, we decided to extend the follow-up period, and the initial draft was used as the basis for the final version two years later. Unfortunately, we overlooked updating the follow-up period in some sections of the text, leading to inconsistencies in the reported timelines. Thank you once again for carefully reviewing the manuscript and pointing out this significant oversight.
Comment 7: figures : add arrows describing area of osteoma on CBCT and x-rays
Response 7: Thank you for your valuable suggestion. Arrows have been added to the radiographs and CBCT scans to highlight the region where the osteoma is located.
Comment 8: was there any further investigation for gardner syndrome ?
Response 8: The possibility that the child might have Gardner's syndrome was explained to the parents. The only available additional diagnostic option for Gardner's syndrome in our region is colonoscopy, which the parents did not consent to. Given that the boy had only one osteoma and no other factors suggesting Gardner's syndrome, it was concluded that the likelihood of the syndrome being present was low. It was agreed that the boy would be clinically monitored at intervals of six months. An explanation of how the patient was monitored for symptoms related to Gardner's syndrome has been added to the case presentation section.
Comment 9: Add potential differential diagnosis in case presentation and discussion
Response 9: We appreciate your helpful suggestion. The potential differential diagnosis has now been incorporated into both the case presentation and discussion sections, offering a more thorough overview of relevant conditions to consider in this case. We trust this revision addresses your concerns and improves the manuscript's clarity.